# Cobalt(II) Paddle-Wheel Complex with 3,5-Di(*tert*-butyl)-4-hydroxybenzoate Bridges: DFT and ab initio Calculations, Magnetic Dilution, and Magnetic Properties

Tatiana V. Astaf'eva [1], Stanislav A. Nikolaevskii [1,*], Evgeniy N. Egorov [2], Stanislav N. Melnikov [1], Dmitriy S. Yambulatov [1], Anna K. Matiukhina [1], Marina E. Nikiforova [1], Maxim A. Shmelev [1], Aleksandr V. Kolchin [1], Nikolay N. Efimov [1], Sergey L. Veber [3], Artem S. Bogomyakov [3], Ekaterina N. Zorina-Tikhonova [1], Igor L. Eremenko [1] and Mikhail A. Kiskin [1,*]

[1] N.S. Kurnakov Institute of General and Inorganic Chemistry, Russian Academy of Sciences, Moscow 119991, Russia; tata1525@mail.ru (T.V.A.); melnikov.stanislav.mail@gmail.com (S.N.M.); yambulatov@yandex.ru (D.S.Y.); matyukhinaanya@gmail.com (A.K.M.); nikiforova.marina@gmail.com (M.E.N.); shmelevma@yandex.ru (M.A.S.); avkolchin432@yandex.ru (A.V.K.); kamphor@mail.ru (E.N.Z.-T.); ilerem@igic.ras.ru (I.L.E.)

[2] Department of Physical Chemistry and Macromolecular Compounds, Chemical-Pharmaceutical Faculty, I.N. Ulianov Chuvash State University, Cheboksary 428015, Russia; enegorov@mail.ru

[3] International Tomography Center, Siberian Branch of the Russian Academy of Sciences, Novosibirsk 630090, Russia; sergey.veber@tomo.nsc.ru (S.L.V.); bus@tomo.nsc.ru (A.S.B.)

* Correspondence: sanikol@igic.ras.ru (S.A.N.); mkiskin@igic.ras.ru (M.A.K.)

**Abstract:** A new binuclear "paddle-wheel" complex, $[Co_2(bhbz)_4(EtOH)_2]\cdot 4EtOH$ (**1**, Hbhbz-3,5-di(*tert*-butyl)-4-hydroxybenzoic acid); an isostructural zinc complex (**2**); a and magnetically diluted sample of $[Zn_{1.93}Co_{0.07}(bhbz)_4(EtOH)_2]\cdot 4EtOH$ (**3**) were obtained. Molecular structures of **1** and **2** were determined by single-crystal X-ray diffraction. DFT calculations for **1** indicate strong Co-Co antiferromagnetic exchange interactions in the binuclear fragment. It was shown that when one paramagnetic ion in the binuclear molecule is replaced by a diamagnetic zinc(II) ion, the remaining cobalt(II) ion can be considered as an isolated center with magnetic anisotropy, the parameters of which are determined by ab initio calculations. Magnetic properties for samples **1** and **3** were investigated and analyzed in detail.

**Keywords:** cobalt carboxylate complexes; magnetic dilution; single-crystal X-ray structure; ab initio calculations; magnetic properties

## 1. Introduction

High interest in coordination compounds based on paramagnetic 3*d* metal ions is caused by the origination of unique magnetic responses, paving the way to a solution of fundamental as well as practical tasks of different complexities [1–6]. The anisotropy of g-tensor parameters, reversible transitions from the low- to the high-spin state of metal centers, as well as exchange interactions between metal centers, stimulating magnetic ordering in the crystal, etc., may be useful for creating memory elements [7–9], optical sensors [10], and soft mechanical actuators [11]. High-spin cobalt(II) ions have maximum magnetic anisotropy in a number of 3*d* metals [12–15], which makes them a promising platform for the effective tuning of axial magnetic anisotropy through a crystal field and obtaining compounds exhibiting slow magnetic relaxation.

The relevance of obtaining multicenter structures promoted the development of single-molecule magnets (SMMs), obtaining structures of different nuclearity from binuclear to 84-nuclear [16], in which exchange interactions between paramagnetic centers can be tuned and slow magnetic relaxation and/or magnetic ordering can be realized [17]. The necessity of adjusting the axial magnetic anisotropy of metal ions determined the direction for the

synthesis of single-ion magnets (SIMs), in which the electronic structure of an ion under the control of a crystal field determines the properties of a molecule in the crystal [17]. Determination of the genuine magnetic properties of isolated molecules in most known SIM examples is unattainable without magnetic dilution which prevents dipole interactions and quantum tunneling of magnetization (QTM), as well as without ab initio calculations for estimating the maximum operating parameters of the molecule.

In most cases, magnetic dilution is performed for single-center cobalt(II) complexes in order to analyze relaxation processes. For example, a complex based on a tetrahedral dianion, $[Co(SPh)_4]^{2-}$, was studied in [18]; it has large negative axial ZFS ($D = -70$ cm$^{-1}$) according to the results of calculations and demonstrates slow relaxation in the absence of an applied field. As the field strength increases, the intensity of one relaxation process (at a higher frequency) decreases while the intensity of the other (at a lower frequency) increases. This reflects the change in relaxation mechanisms from thermal activation (at higher frequencies) to quantum tunneling (at lower frequencies) depending on the magnitude of the applied direct current (DC) magnetic field. The study of relaxation mechanisms for an isomorphic magnetically diluted ZnCo sample showed that the second relaxation process—that is, the evidence of the intermolecular nature of the second process observed in the original sample—was not observed. In the investigation performed by Zadrozny and co-authors [19], it was shown that the magnetic dilution of the mononuclear complex $K(Ph_4P)[Co(OPh)_4]$ "switched off" the intermolecular exchange interactions which prevented the observation of thermally activated magnetic relaxation. The magnetic dilution of the complex $[Co(L)(OAc)Y(NO_3)_2]$ (where L is N,N′,N″-trimethyl-N,N″-bis(2-hydroxy-3-methoxy-5-methylbenzyl)-diethylenetriamine) [20] showed that the observed slow magnetic relaxation is realized with the participation of optical/acoustic Raman processes since the dependence of the relaxation time is determined by the law $T^{-n}$ at $n = 4.5$, whereas Raman relaxation for the Kramers ion is expected only at $n = 9$. Magnetic dilution can also lead to a change in the coordination geometry of cobalt ions, which was studied in detail in the example of a mononuclear complex. The sample changed the environment of cobalt ions from $CoN_2O_4$ (a distorted octahedron) to $CoN_2O_2$ (a distorted tetrahedron), which was caused by increasing Co-O distances during the transition to a diamagnetic sample; this was accompanied by a change in the sign of magnetic anisotropy and a switching of relaxation mechanisms [21]. Recently, the 3D polymer $(CH_6N_3)[Co(HCOO)_3]$ has been studied, and for it, spin-canted antiferromagnetism, together with hysteresis below $T_N = 14$ K, has been revealed [22]. In this work, it was shown that a similar diamagnetic matrix, $(CH_6N_3)[Zn(HCOO)_3]$, can be used for Co(II) ion doping, manifesting itself as an SIM with a positive $D$ term of zero-field splitting.

Carboxylates of $3d$ metals are being actively investigated for various applications. It is worth noting that the ability to adjust coordination modes in carboxylate complexes allows for varying the parameters of exchange interactions. The tetracarboxylate dimer of copper(II) acetate in which strong antiferromagnetic interactions are realized is a classic example of spin-coupled systems. Similar dimers of chromium(III), manganese(II), iron(II), and nickel(II) ions have also been studied. The geometry of the coordination environment of the metal atom in such compounds corresponds to a square pyramid or a vacant octahedron. Mononuclear complexes in which planar or axial magnetic anisotropy can be realized depending on the distortion of the polyhedron are known for Co(II) coordination compounds. The latter fact formally makes these objects promising for SMM design. Since we expect the occurrence of strong antiferromagnetic spin–spin exchange interactions between ions inside the binuclear molecule $LCo(\mu-O_2CR)_4CoL$, replacing one of the cobalt atoms with a diamagnetic M(II) ion (for example, zinc) can allow us to observe the magnetic properties of individual cobalt ions.

Substituents at carboxylate groups in such molecules can be used as a tool for controlling the composition and structure of the complex. At the same time, bulk substituents R can help to shield the paramagnetic centers of molecules from each other.

Previously, we prepared the nickel(II) binuclear complex [Ni$_2$(bbz)$_4$(2,3-lut)$_2$] [23] with sterically demanding anions of 3,5-di-*tert*-butyl benzoic acid (Hbbz), and the trinuclear cobalt complex [Co$_3$(bbz)$_6$(EtOH)$_2$] [24]. In order to obtain a binuclear cobalt complex with a bulk anion of a similar structure, we used 3,5-di(*tert*-butyl)-4-hydroxybenzoic acid (hbhbz). As a result, the binuclear complex [Co$_2$(bhbz)$_4$(EtOH)$_2$]·4EtOH (**1**), which is the object of consideration in this work, was synthesized. To study the magnetic characteristics of cobalt(II) ions in **1** (taking into account magnetic dilution), we synthesized an isostructural diamagnetic complex based on Zn(II) ions, [Zn$_2$(bhbz)$_4$(EtOH)$_2$]·4EtOH (**2**), and a congener in which Zn(II) ions are partially replaced by "magnetic" Co(II) ions, [Zn$_{1.93}$Co$_{0.07}$(bhbz)$_4$(EtOH)$_2$]·4EtOH (**3**).

It is worth noting that compounds based on 3,5-di(*tert*-butyl)-4-hydroxybenzoic acid exhibit redox-active behavior. Firstly, a catalytically promoted oxidation of Hbhbz to 2,6-di-*tert*-butylhydroquinone with cobalt(II) and nickel(II) salts followed by decarboxylation and recombination into diquinone was previously shown [23,25] (Scheme 1). Secondly, the presence of an oxidant can generate a stable radical from bhbz, which was previously shown on a nickel(II) complex with a coordinated bhbz anion [26]. These factors needed to be taken into account during the synthesis of **1–3**, and the reactions were carried out in an inert atmosphere at room temperature.

**Scheme 1.** The formation of 3,3′,5,5′-tetra-*tert*-butyl-1,1′-biphenylidene-4,4′-quinone from two molecules of 4-hydroxy-3,5-di-*tert*-butylbenzoic acid.

## 2. Results

### 2.1. Synthesis and Characterization

The binuclear complexes [Co$_2$(bhbz)$_4$(EtOH)$_2$]·4EtOH (**1**) and [Zn$_2$(bhbz)$_4$(EtOH)$_2$]·4EtOH (**2**) were synthesized by the interaction of corresponding metal chloride (Co$^{2+}$ or Zn$^{2+}$) and potassium 3,5-di(*tert*-butyl)-4-hydroxybenzoate (Kbhbz) in a 1:2 molar ratio in ethanol (Scheme 2). X-ray-quality single crystals of **1** (blue) and **2** (colorless) were isolated directly from the mother liquor. According to XRD, compounds **1** and **2** are isostructural. After isolation as crystalline solids, both compounds **1** and **2** were found to be poorly soluble in ethanol. Recrystallization of **1** and **2** from acetonitrile led to a precipitation of polycrystalline products. Recrystallization of **2** from benzene resulted in the isolation of a new oxo-carboxylate tetranuclear complex, [Zn$_4$O(bhbz)$_6$]. Based on the above reasons, the synthesis of the diamagnetically diluted sample was carried out using the procedure described for the synthesis of **1** and **2** but with the metal ratio Co:Zn as 1:19. A polycrystalline product (blue) was isolated in two steps: the first portion (**3a**) in 24 h and the second one (**3b**) in 72 h. According to inductively coupled plasma optical emission spectroscopy (ICP-OES), the cobalt content in **3a** corresponds to 3.6%, and that in **3b** corresponds to 3.9%.

After isolation of **3b**, the solution was kept in air, and two types of crystals (colorless and dark red) were isolated additionally. According to XRD, colorless crystals were complex **2**, and dark red crystals were the product of oxidation/decarboxylation of bhbz acid and the formation of diquinone (Scheme 1).

**Scheme 2.** The formation of complexes **1** and **2**.

The obtained IR spectra of **1**–**3** (Figure S1) contained stretching vibrations (in cm$^{-1}$): C-H ($\nu_{as}$(C(CH$_3$)$_3$) at 2959; $\nu_{as}$(C-H$_{ring}$) at 2912 and 2876; $\delta_{as}$(CH$_3$) at 1452; $\delta_s$(C(CH$_3$)$_3$) at 1323; $\gamma$(C(CH$_3$)$_3$) at 1202; $\delta$(C-H$_{ring}$) at 920, 890, and 695), C-C ($\nu$(C-C$_{ring}$) at 1634 and 1593), C-O ($\nu_{as}$(COO) at 1550; $\nu_s$(COO) at 1392; $\nu$(PhC-O) at 1241; $\delta$(COO) at 787, 660, and 614; $\rho$(COO) at 552), O-H ($\nu$(O-H) at 3630) [27].

Intense bands of carboxylic groups are observed in Raman spectra of **1**–**3**: 705, 931, 934 cm$^{-1}$ ($\delta$(C-H$_{ring}$)); 552 ($\rho$(COO)); and 755 cm$^{-1}$ ($\delta$(COO)) (Figure S2). Low-intensity bands were found in the spectrum, suggesting the presence of cobalt (Co$_3$(HCOO)$_6$. [28]) and/or zinc formates (Zn$_3$(HCOO)$_6$ [29] as an impurity in the samples. Low-intensity bands at 780 cm$^{-1}$ for **2** and **3** and 790 cm$^{-1}$ for **1** and **3** can be attributed to $\delta_s$(COO$^-$) for Zn$_3$(HCOO)$_6$ and Co$_3$(HCOO)$_6$, respectively, and the band at ~1060 cm$^{-1}$ corresponds to $\delta_{as}$(CH) [30].

According to the XPRD data for **1**–**3** in the 2$\theta$ range from 5 to 45 deg (Figures S3–S5), the samples are single-phased, and the slight background in the diffractograms may be due to some amorphous impurity.

## 2.2. Crystal Structures

Compound **1** crystallizes as a binuclear complex, [Co$_2$(bhbz)$_4$(EtOH)$_2$], with four solvated ethanol molecules. The molecule of [Co$_2$(bhbz)$_4$(EtOH)$_2$] is centrosymmetric; the metal atoms are linked by four bridging carboxylate groups of bhbz (Co...Co 2.7266(9) Å) (Figure 1). The cobalt atom is surrounded by five oxygen atoms, and the geometry of the CoO$_5$ environment corresponds to a square pyramid ($\tau$ = 0). The oxygen atoms of carboxylate groups are located at the base of the pyramid (Co-O 2.018(2)-2.032(2) Å), the cobalt atom deviates from the O$_4$ plane by 0.2487(11) Å, and the axial position is occupied by the O atom of the coordinated EtOH molecule (Co-O 2.038(2) Å). The molecular structure of the dimeric complex is stabilized by the intramolecular C-H...O contacts (Table S1).

In the crystal, dimeric molecules form a supramolecular chain (Co...Co$_{min}$ 12.469 Å) along the 0*b* axis through the C-H...$\pi$ interaction between the methyl proton of the *tert*-butyl group and the benzene ring of the carboxylate anion (Figure 2, Table S2). Along the axes, the crystal packing is augmented by the intermolecular O-H...O interactions of solvated ethanol molecules as well as by C-H...$\pi$ contacts (Figure 2, Tables S1 and S2). The Co1Co1 vector of the dimer fragment is co-directional to the 0*a* axis, and a chain of dimers (Co...Co$_{min}$ 8.945 Å) contacting through the H-bonds of coordinated and solvated ethanol molecules is lined up along it (Figure 2, Table S1). The nearest distance between the metal atoms of adjacent molecules located along the 0*c* axis and interacting only due to van der Waals contacts is 12.988 Å (Figure 2).

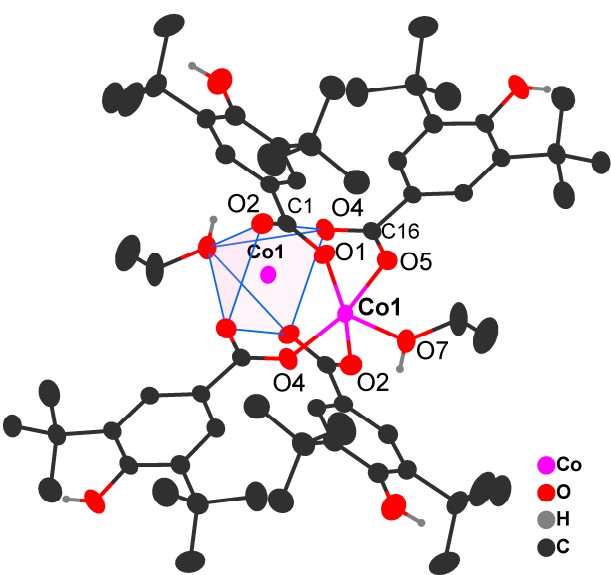

**Figure 1.** Molecular structure of [Co$_2$(bhbz)$_4$(EtOH)$_2$] in **1** (H atoms at carbon atoms are not shown; ellipsoids are shown with a probability level of 30%).

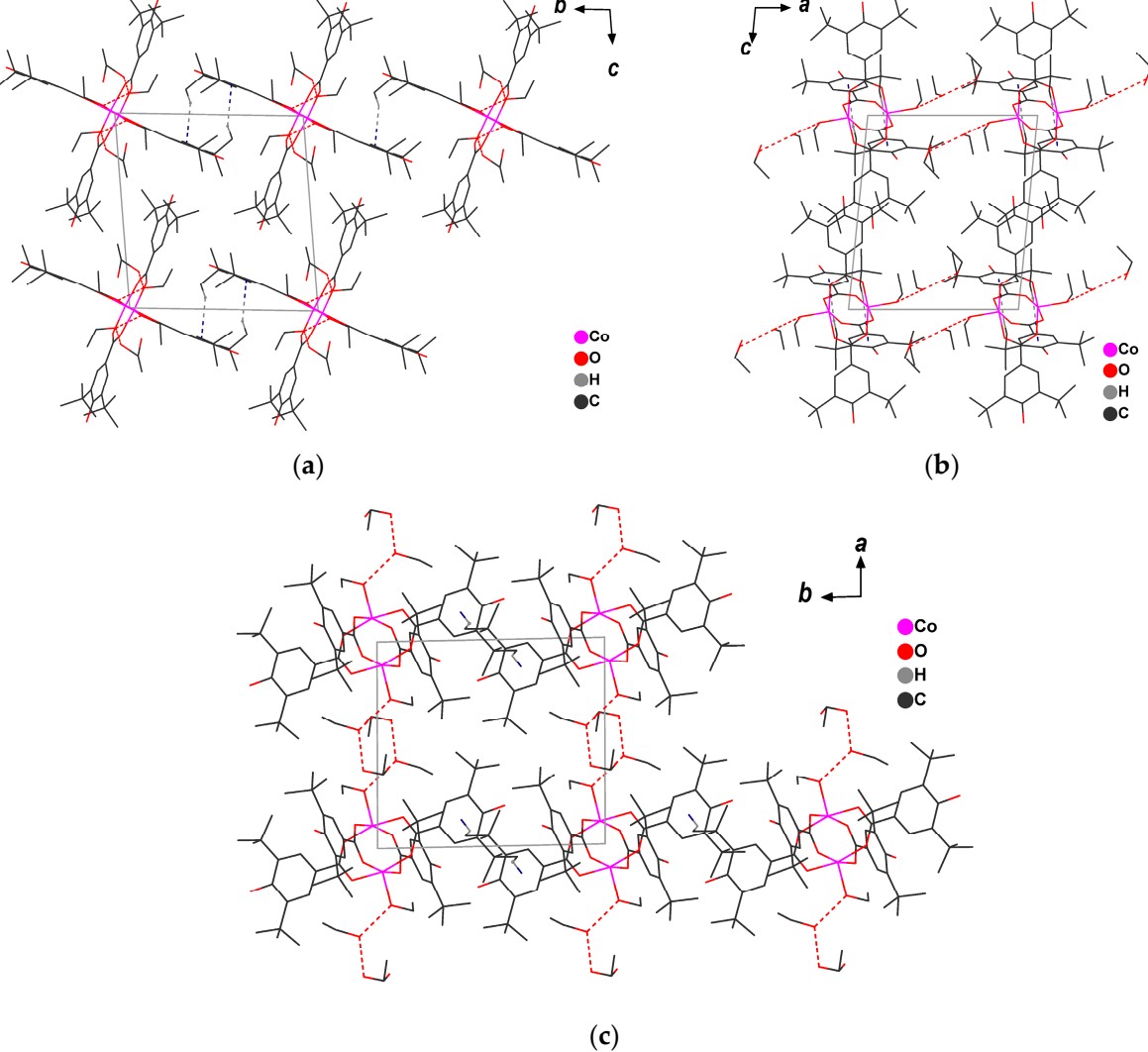

**Figure 2.** Crystal packing of **1** ((**a**) projection onto plane *bc*, (**b**) projection onto plane *ac*, (**c**) projection onto plane *ab*).

Compound **2** is isostructural to **1**. Close values of interatomic distances and angles in **1** and **2** are observed. The intramolecular distance of Zn...Zn is 2.8784(6) Å, Zn-O(O$_2$CR) bond lengths are in the range of 2.0300(11)-2.0436(11) Å, Zn-O(EtOH) is 1.9905(11) Å, and the zinc atom deviates from the plane of the pyramid base by 0.3240(6) Å. The transition from a Co- to Zn-containing complex is accompanied by minor changes in the structure of the molecule [M$_2$(bhbz)$_4$(EtOH)$_2$]: increasing the distance M...M by 0.151 Å causes displacement of the zinc atom closer to the geometrical center of the square pyramid (by 0.076 Å) and is partially compensated by the proximity of the oxygen atom of the ethanol molecule (by 0.049 Å). This change in the geometry of the coordination environment can be expressed by the square pyramid distortion criterion defined by the Continuous Shape Measures approach: $S_Q(1) = 0.465$ and $S_Q(2) = 0.326$.

The observed M-O and C-O bond lengths agree well with the known data for dimeric Co and Zn complexes of similar structures [31–34].

### 2.3. Calculations

#### 2.3.1. BS/DFT

To assess the strength of exchange interactions of Co$^{2+}$ ions in binuclear complex **1**, density functional theory (DFT) was applied to calculate the exchange coupling constant J. Broken-symmetry calculations were performed on molecular geometries extracted from the X-ray data at the B3LYP/def2-TZVP level of theory (ESI, Table S3) [35–37]. Assuming that the dominant magnetic exchange between adjacent Co ions in the structure occurs through oxygen atoms belonging to carboxylate bridges, the following Heisenberg–Dirac–Van Vleck Hamiltonian was applied [38,39],

$$\hat{H} = -2J_{Co1-Co2}\left[\hat{S}_{Co1}\hat{S}_{Co2}\right] \tag{1}$$

where $\hat{S}$ is the total spin momentum operator and $J_{Co1-Co2}$ is a parameter corresponding to Co-Co exchange coupling.

Since the intermetallic distance is quite small, a significant overlap of local magnetic orbitals is possible in the complex, for which the Yamaguchi [40] interaction theory was used (ESI). The resulting parameter $J = -62.73$ cm$^{-1}$ is negative, which indicates the presence of strong antiferromagnetic interactions in **1**.

#### 2.3.2. Ab Initio

For theoretical interpretation and for proof that the absence of relaxation in the case of **1** is caused precisely by antiferromagnetic exchange, ab initio multiconfigurational calculations were carried out using the CASSCF/NEVPT2/def2-TZVP level. Calculations were also carried out for the Co$^{2+}$ ion in the diamagnetically diluted complex **3**, i.e., a "Zn-matrix". The geometry of the cobalt(II) environment in **1** and **3** was taken from X-ray data for **1** and **2**, respectively. The symmetry of the point group for polyhedrons in both complexes corresponds to C$_{4v}$ with the distortion parameters 0.022 for **1** and 0.032 for **3**.

For the d$^7$ configuration, the terms in spherical symmetry are $^4$F. With the application of a C$_{4v}$ crystal field, the term splits into $^4$T$_2$, $^4$T$_1$, and $^4$A$_2$ [41]. Next, the $^4$T$_2$ triplet splits into a $^4$E doublet and a $^4$A$_2$ singlet. In the case of the lowest energy, the ground state is a singlet, and the spin Hamiltonian formalism is applicable. However, for tetragonal-pyramidal complexes, there may be a contribution from the unquenched orbital momentum [42–44], since at the electronic level it is possible to consider both for an elongated bipyramid [45]. In this case, the ground state is an orbital doublet, and it is necessary to operate with the Griffith–Figgis Hamiltonian formalism [46],

$$H = -\frac{3}{2}\kappa\lambda LS + \Delta_{ax}\left[L - \frac{1}{3}L(L+1)\right] + \Delta_{rh}(\hat{L}_X^2 - \hat{L}_Y^2) + \mu_B B\left(g_e\hat{S} - \frac{3}{2}\kappa L\right) \tag{2}$$

where $\kappa$ is an orbital reduction factor (often 0.7–1), $\lambda$ is a spin-orbit splitting parameter, $L = 1$ (due to T-P isomorphism), $\hat{L}$ is a vector operator of angular momentum and its components ($x$, $y$, $z$), and $\Delta_{ax}$ and $\Delta_{rh}$ are axial and rhombic parameters of a crystal field.

The calculated energies of the main terms for complexes **1** and **3** are given in Table 1. In the case of **1**, the calculated parameter $\Delta_{ax} > 0$, which makes it possible to use the spin Hamiltonian to interpret the dc-magnetic behavior. The axial parameter of ZFS is 90.288 cm$^{-1}$, and "easy-plane" anisotropy with close values is quite typical for cobalt complexes with SPY or vOC geometry of the coordination environment [47–55]. In contrast, for the Co$^{2+}$ ion in Zn-diluted complex, the ground crystal-field term is double degenerate and $\Delta_{ax} < 0$. However, the effective Hamiltonian-calculated parameters show that $D$ for this complex must be positive. The $E/D$ ratio for this complex is quite high (0.289), which raises questions about the genuine nature of the $D$ sign and indicates the implementation of triaxial anisotropy [43,56–59].

**Table 1.** CASSF/NEVPT2-calculated values of parameters of the spin Hamiltonian and the Griffith–Figgis Hamiltonian for the ground term.

|  | **1** | | **3** | |
|---|---|---|---|---|
| Initial States (cm$^{-1}$) | $^4A_2$ | 0 | $^4E$ | 0<br>395.6 |
|  | $^4E$ | 679<br>1296.9 | $^4A^2$ | 1117.8 |
| SH parameters from effective Hamiltonian for the ground term | | | | |
| $D$ (cm$^{-1}$) | 90.288 | | 100.565 | |
| $E/D$ | 0.203 | | 0.269 | |
| $g_x$ | 1.831 | | 1.746 | |
| $g_y$ | 2.625 | | 2.589 | |
| $g_z$ | 2.983 | | 3.102 | |
| $g_{iso}$ | 2.48 | | 2.479 | |
| GF Hamiltonian | | | | |
| $\Delta_{ax}$ (cm$^{-1}$) | 987.95 | | −920.0 | |
| $\Delta_{rh}$ (cm$^{-1}$) | 308.95 | | \|197.8\| | |
| $\lambda$ (cm$^{-1}$) | −173.41 | | $\lambda = -173.331$ | |

At the same time, it is possible to estimate the energies of the Kramers doublets (KDs) quite accurately, since they depend notably on the splitting of terms as the spherical symmetry decreases. Comparative analysis of KD energies (Table 2) indicates a significant change in the electronic structure of Co$^{2+}$ ions upon Zn dilution of complex **1**.

**Table 2.** CASSCF/NEVPT2-calculated values of the six lowest Kramers doublets (cm$^{-1}$).

| **1** | **3** |
|---|---|
| 0 | 0 |
| 191.4 | 221.9 |
| 953.5 | 736.5 |
| 1150.6 | 986.1 |
| 1636.6 | 1523.9 |
| 1697.4 | 1594.2 |

For **1** and **3**, the first excited KD$^{I}$ is located at a distance exceeding the phonon energy, which excludes the possibility of implementing the classical Orbach process. In the first complex, the energy difference between KD$^{I}$ and KD$^{II}$ is 762.1 cm$^{-1}$, which implies that the magnetic behavior for one Co$^{2+}$ ion is determined exclusively by the orbital singlet. For the second complex, a negative value of $\Delta_{ax}$ means that the excited orbital singlet $^4A_2$ does not affect the magnetic data, which are determined exclusively by the doublet $^4E$ [60]. In a

first approximation, the probabilities for the implementation of relaxation mechanisms are close for each of the $Co^{2+}$ ions. It is also possible for **3** to compete with direct/Raman and TA-QTM (Figure S6). In other words, the absence of slow magnetic relaxation for the first complex is determined by the presence of AF exchange.

Also, for a complete view of the electronic structure of the ions in both cases, the energies of the *d*-orbitals were calculated using ab initio ligand field theory (AILFT) (Figure 3).

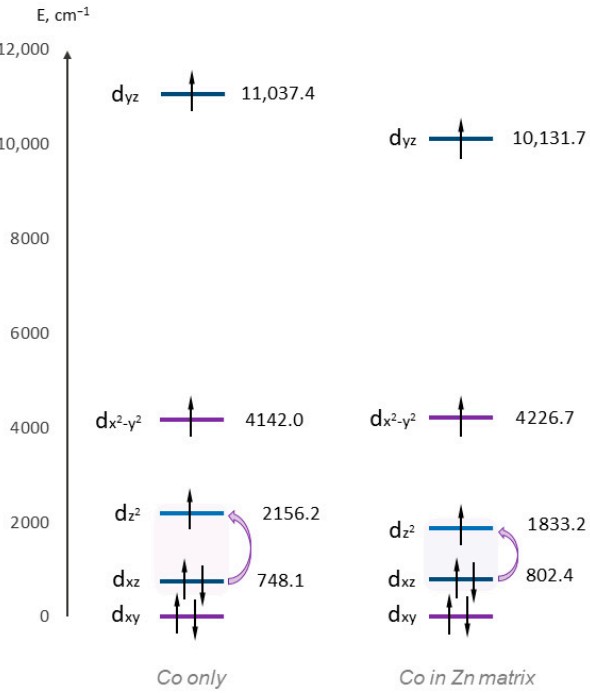

**Figure 3.** CASCCF/NEVPT2 relative energies of *d*-orbitals. The violet arrows indicate the lowest energy transitions.

The last doubly occupied orbital is $d_{xz}$, and the first half-occupied orbital for both complexes is $d_z^2$. Since these orbitals have different $|m_l|$ values, the contribution to the value of the axial ZFS parameter *D* must be positive [53,61,62]. The energy gap for complex **1** is 1408.1 cm$^{-1}$, while in the case of **3**, the transition energy is only 1030.8 cm$^{-1}$, which explains the higher *D* values for the first complex.

*2.4. Magnetic Properties*

The results of the study of magnetic properties of **1** are presented in Figure 4. The value of $\chi T$ at 300 K is 4.57 cm$^3$·K·mol$^{-1}$, which is significantly lower than the theoretical value of 6.85 cm$^3$·K·mol$^{-1}$ for two noninteracting Co(II) ions with spin $S = 3/2$. Upon temperature lowering, $\chi T$ decreases, reaching a small plateau at ~0.16–0.18 cm$^3$·K·mol$^{-1}$ below 25 K, which indicates the presence of strong antiferromagnetic exchange interactions leading to the compensation of spins in the binuclear molecule and, as a consequence, to the diamagnetic ground state. Non-zero values of $\chi T$ below 25 K are associated with the presence of a small amount of paramagnetic impurity ($p < 3\%$). The repeated studies of the magnetic susceptibility of the newly obtained sample **1** confirmed the reproducibility of the observed magnetic data. The presence of a significant orbital contribution to the magnetic susceptibility (that is common for Co(II) ions) complicates the analysis of experimental data. Simultaneous consideration of spin-orbital and exchange interactions leads to overparameterization, which prevents correct values of the energy of exchange interactions in the dimer from being obtained.

EPR spectroscopy at 10 K shows that sample **1** does not possess the X-band signal. This fact confirms the diamagnetic ground state of the dimer ($S = 0$), i.e., the performance of

strong exchange interactions between paramagnetic centers (see section BS/DFT), leading to a complete pairing of the spins.

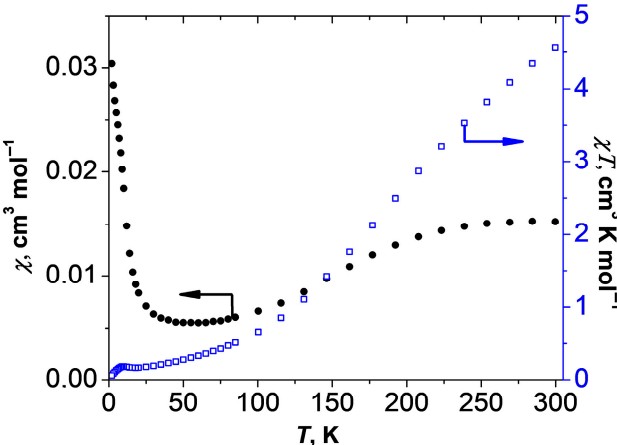

**Figure 4.** Temperature dependences of magnetic susceptibility (●) and $\chi T$ of **1** ($H$ = 5 κOe).

The magnetic behavior of the diamagnetically diluted sample **3** is strongly different from that of binuclear complex **1**. The $\chi T$ value of **3** at room temperature is 0.35 cm$^3$·K·mol$^{-1}$, which corresponds to the cobalt(II) ion content of ~3% in the sample. As the temperature decreases, the value of $\chi T$ decreases gradually to 0.17 cm$^3$·K·mol$^{-1}$ at 2 K (Figure 5). Analysis of the $\chi T(T)$ dependence in the temperature range 2–130 K allows us to estimate the magnetic anisotropy parameter $D$. The optimal value of $D$ obtained by analysis in the framework of the spin Hamiltonian (3) taking into account the correction for temperature-independent paramagnetism (*TIP*) is 75.0 cm$^{-1}$ at $g_{iso}$ = 2.5 and *TIP* = 1.15 × 10$^{-4}$ ($R_2$ = 6.7 × 10$^{-5}$).

$$\hat{H} = D\hat{S}^2 + g_{iso}\beta\hat{S}H \tag{3}$$

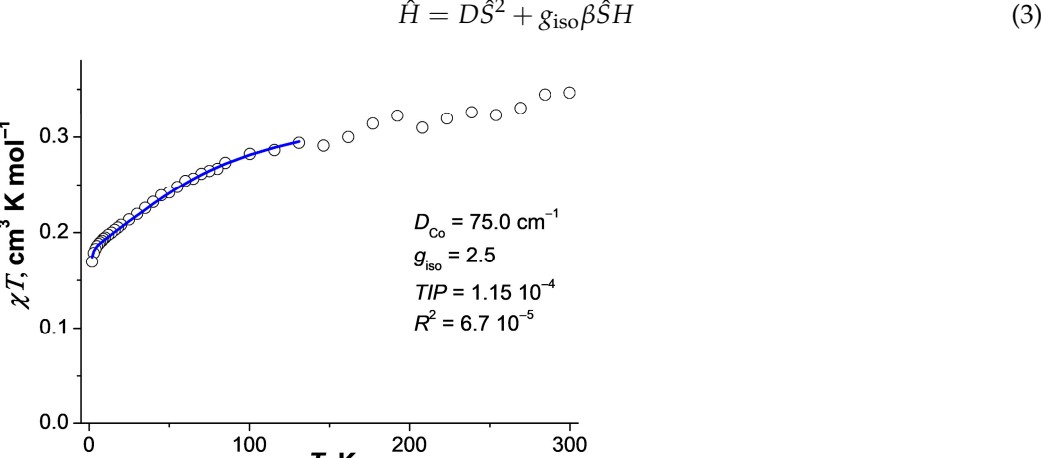

$D_{Co}$ = 75.0 cm$^{-1}$
$g_{iso}$ = 2.5
*TIP* = 1.15 10$^{-4}$
$R^2$ = 6.7 10$^{-5}$

**Figure 5.** Temperature dependence of $\chi T$ for **3** ($H$ = 5 kOe). Blue line is simulation of experimental data in the temperature range 2–130 K (see main text).

The field dependence of the magnetization for complex 3 (Figure S7) attained saturation with ~0.22 μB magnetization values. This fact is consistent with the presumable level of Co(II) content in sample **3**, about 3.7%.

The X-band EPR spectroscopic studies for **3** at 10 K recorded a signal (Figure S8) that was simulated using EasySpin software. The EPR spectrum of complex **3** is typical for high-spin Co(II) ions; the width of the line is 6 mT. The simulated spectrum at the effective spin of the system $S$ = 1/2 allowed us to determine the Hamiltonian parameters: $g_x = g_y$ = 4.6; $g_z$ = 2.11. The Hamiltonian parameters for the model of effective spin $S$ = 3/2 are as follows: $g_x = g_y$ = 2.29; $g_z$ = 2.12. The estimated value of the zero-field splitting parameter $D$ of the cobalt(II) ion used in the simulations was about 23 cm$^{-1}$. The parameter

$D$ has a positive sign; i.e., the state with spin $\pm 1/2$ is the ground state. It is not possible to refine the rhombicity parameter $E$.

AC measurements in a zero external magnetic field for **3** showed the growth of the frequency dependence of the imaginary part of dynamic magnetic susceptibility at 2 K, but no maxima were observed. This indicates the presence of slow magnetic relaxation but does not allow us to quantify the relaxation processes in the complex. In order to suppress the possible influence of the quantum tunneling of magnetization (QTM), we varied the strength of the external magnetic field. The optimal value of the magnetic field strength at which the relaxation time for a given sample is maximal is 1000 Oe (Figures S9 and S10) for **3** (the maximum of the frequency dependence of the imaginary part of dynamic magnetic susceptibility is located at the minimum possible frequency). Measurements of isotherms of frequency dependences of dynamic magnetic susceptibility in an optimal external DC magnetic field and approximation of $\chi''$ ($\nu$) using the generalized Debye model (Figure 6) allow us to deduce dependences of the relaxation time from the inverse temperature $\tau(1/T)$ (Figure 7). The dependence $\tau(1/T)$ over the investigated temperature range for **3** is approximated by the Raman ($\tau_{Raman}^{-1} = C_{Raman} \cdot T^n$) mechanism: $C_{Raman} = 0.038 \pm 0.020 \ K^{-n}s^{-1}$; $n = 7.6 \pm 0.3$. The value of the Raman exponent is close to the common value for a Kramers ion such as Co(II), $n = 7$, and the $C_{Raman}$ value is in the expected range of $10^{-5}$ to $10^{-1}$ [63]. The Cole–Cole plots imply a single relaxation magnetic center (Figure S11).

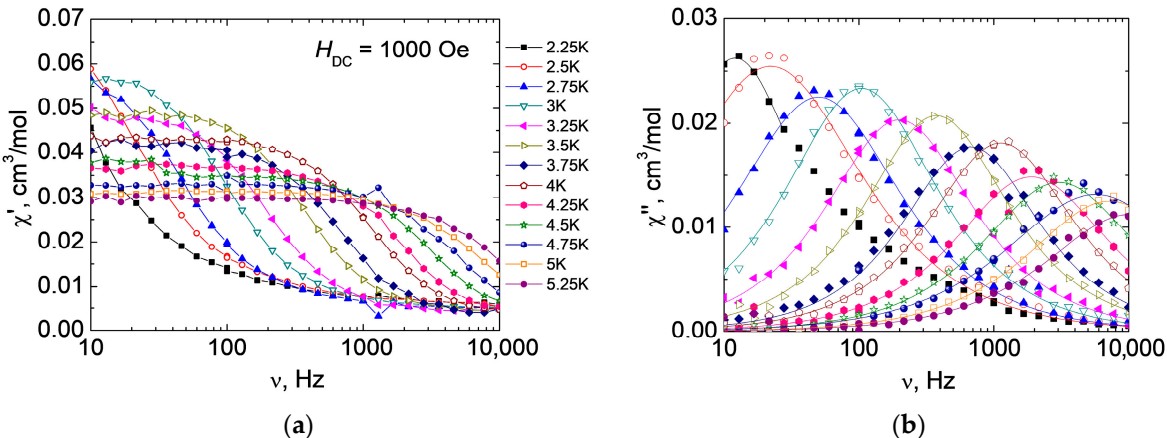

**Figure 6.** Frequency dependences of the real ($\chi'$) (**a**) and imaginary ($\chi''$) (**b**) parts of the dynamic magnetic susceptibility of **3** at 2.25–5.25 K in 1000 Oe external DC magnetic field. Solid lines—approximation of experimental data using the generalized Debye model.

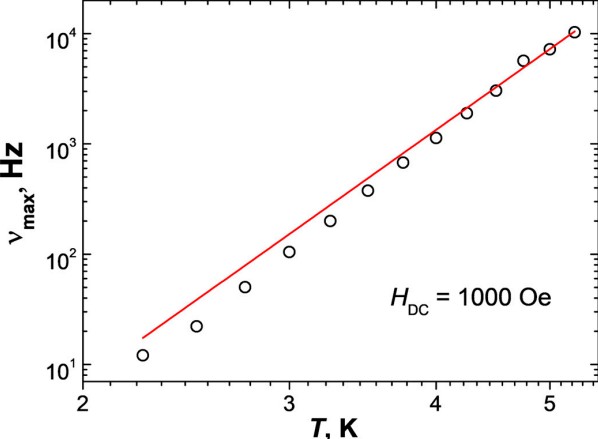

**Figure 7.** Temperature dependence of relaxation frequency for **3** in 1000 Oe dc-field. Red solid line—approximation using the Raman mechanism.

## 3. Discussion

Initially, we designed this investigation aiming to reveal the effect of the diamagnetic dilution of the binuclear paddle-wheel type complex **1** on its magnetic properties. During the investigation, we were surprised by the following features of the object under study: (i) A low solubility of the complex **1** in ethanol complicated its possible purification by recrystallization. (ii) The oxidation of the bhbz anion to 3,3′,5,5′-tetra-*tert*-butyl-1,1′-biphenylidene-4,4′-quinone in the presence of metal ions opened one more way for sample contamination. The quinone has a good solubility in ethanol and in all cases crystallized after isolation of crystals **1**–**3** by prolonged storage of the mother liquor. The most serious problem for the isolation of desirable complexes as a pure phase is the oxidation co-product (presumably formate), which according to the spectral and magnetochemical data (for **1**) can co-crystallize with **1**–**3** as an impurity. Cobalt(II) formates are three-dimensional coordination polymers ($D_{calc}$: ~1.8–1.9 g·cm$^{-3}$ vs. 1.2 g·cm$^{-3}$ for **1** and **2**) exhibiting spin-canted antiferromagnetism or weak ferromagnetic properties [64]. An impurity of such a magnetic substance with a high concentration of paramagnetic centers ($C_{Co}$ is close to ~26% for cobalt formate vs. 9% for **1**) can make an accurate interpretation of magnetic data impossible for the initial complex **1** and the magnetically diluted sample. The specific intensity of the cobalt formate magnetization is higher than that of **1,** which is consistent with problems that we faced in performing the approximation of the $\chi T(T)$ dependences.

## 4. Materials and Methods

### 4.1. Instruments

Synthetic manipulations were performed using the standard Schlenk technique. All reagents were commercially available and were used without further purification (ZnCl$_2$, CoCl$_2$·6H$_2$O, Hbhbz, KOH, EtOH). The IR spectra of the compounds were recorded on a Perkin Elmer Spectrum 65 spectrophotometer equipped with a Quest ATR Accessory (Specac) with attenuated total reflectance (ATR) in the range of 400-4000 cm$^{-1}$. The Raman spectra were acquired using a SOL Instruments Confotec NR500 (excitation wavelength, 785 nm; radiation power, 7.5 mW; the accumulation time and the number of counts were 5 s and 5, respectively) with an Olympus PLN 20X Objective (Numerical Aperture 0.4). Elemental analysis was performed on an automatic EuroEA-3000 C, H, N, S analyzer (EuroVektor). The cobalt and zinc concentrations in the samples were determined by optical emission spectroscopy with inductive-coupled plasma (ICP-OES), using a Thermo Scientific iCAP XP spectrometer. X-ray powder diffraction (XRPD) patterns were recorded with a Bruker D8 Advance diffractometer using Cu$_{K\alpha}$ radiation ($\lambda$ = 1.5406 Å) within a 2θ range of 5–45° and with a signal collection time of 0.1 s per step. Magnetic behavior was studied using a Quantum Design PPMS-9 system. The temperature dependences of the magnetization (*M*) were measured in a 0.5 T magnetic field in the temperature range of 2–300 K during cooling. During AC susceptibility measurements in the frequency range of 10–10$^5$ Hz, the alternating magnetic field amplitude was $H_{ac}$ = 1–5 Oe. The measurements were carried out on samples that were moistened with mineral oil to prevent any texturizing of the particles in the DC magnetic field. The prepared samples were sealed in polyethylene bags. The magnetic susceptibility $\chi = M/H$ was determined taking into account the contribution of the bag and that of the mineral oil. The obtained data for **3** were recalculated per cobalt atom according to ICP-OES results. EPR spectra were recorded using Bruker Elexsys E580 spectrometer at the X-band (9 GHz) in continuous wave mode. The spectrometer was equipped with an Oxford Instruments temperature control system, and powder polycrystalline spectra were measured at *T* = 10 K.

### 4.2. Synthesis

General procedure. Potassium 3,5-di-*tert*-butyl-4-hydroxybenzoate Kbhbz was generated in situ by the reaction of KOH (112 mg, 2 mmol) and Hbhbz (500 mg, 2 mmol) in EtOH (15 mL). The solution of Kbhbz was added to a solution of CoCl$_2$·6H$_2$O or anhydrous ZnCl$_2$ (1 mmol) in EtOH (25 mL) in an inert atmosphere. The reaction mixture was kept at

room temperature for 30 min upon stirring, and then it was filtered from KCl. The residual solution was kept at room temperature in an inert atmosphere.

[Co$_2$(bhbz)$_4$(EtOH)$_2$]·4EtOH (**1**). CoCl$_2$·6H$_2$O (238 mg, 1 mmol), potassium hydroxide KOH (112 mg, 2 mmol), Hbhbz (500 mg, 2 mmol), and EtOH (40 mL) were used. The violet solution was kept at room temperature under vacuum. Blue single crystals obtained in 24 h were decanted, washed with cold (5 °C) EtOH, and dried under ambient conditions. The yield was 1.02 g (73%). Anal. Calc. for C$_{72}$H$_{120}$O$_{18}$Co$_2$: C, 62.43; H, 8.71. Found: C, 62.16; H, 8.63.

[Zn$_2$(bhbz)$_4$(EtOH)$_2$]·4EtOH (**2**). Anhydrous ZnCl$_2$ (136 mg, 1 mmol), potassium hydroxide KOH (112 mg, 2 mmol), Hbhbz (500 mg, 2 mmol), and EtOH (40 mL) were used. The yellow solution was kept at room temperature under vacuum. Single crystals obtained in 24 h were decanted, washed with cold (5 °C) EtOH, and dried under ambient conditions. The yield was 1.08 g (77%). Anal. Calc. for C$_{72}$H$_{120}$O$_{18}$Zn$_2$: 61.62; H, 8.56. Found: C, 61.89; H, 8.51.

[Co$_{1.93}$Zn$_{0.07}$(bhbz)$_4$(EtOH)$_2$]·4EtOH (**3**). Two solutions were prepared under vacuum. The first one was prepared from anhydrous ZnCl$_2$ (258 mg, 1.9 mmol), KOH (202 mg, 3.8 mmol), Hbhbz (950 mg, 3.8 mmol), and EtOH (40 mL). The second one was prepared from CoCl$_2$·6H$_2$O (24 mg, 0.1 mmol), KOH (11 mg, 0.2 mmol), Hbhbz (50 mg, 0.2 mmol), and EtOH (20 mL). The solutions were mixed and then kept at room temperature under vacuum. Violet single crystals obtained in 24 h were decanted, washed with cold (5 °C) EtOH, and dried under ambient conditions. The cobalt–zinc ratio in **3** was detected by ICP-OES.

### 4.3. Single-Crystal X-ray Diffraction

Single-crystal X-ray studies of crystals **1** and **2** were carried out on a Bruker APEX II diffractometer with a CCD detector (MoK$_\alpha$, $\lambda$ = 0.71073 Å, graphite monochromator) [65]. A semiempirical adjustment for absorption was introduced for **1** and **2** [66]. The structures were solved with direct methods and refined by the least-squares method in the full-matrix anisotropic approximation on $F^2$. All hydrogen atoms were located in calculated positions and refined within the riding model. All calculations were performed using the SHELXTL [67,68] and Olex2 [69] software packages. The structures were solved using restraints (DFIX, ISOR) and taking into account the disorder of solvent molecules. The crystallographic parameters and the structure refinement statistics are shown in Table 3. Supplementary crystallographic data for the compounds synthesized are given in CCDC numbers 2320605 (for **1**) and 2320606 (for **2**). These data can be obtained free of charge from The Cambridge Crystallographic Data Centre via www.ccdc.cam.ac.uk/data_request/cif (accessed on 22 December 2023).

**Table 3.** Selected crystal data and parameters for structure refinement of **1** and **2**.

| Complex/Parameters | 1 | 2 |
|---|---|---|
| Empirical formula | C$_{72}$H$_{120}$Co$_2$O$_{18}$ | C$_{72}$H$_{120}$O$_{18}$Zn$_2$ |
| Formula weight | 1391.53 | 1404.41 |
| $T$ (K) | 150(2) | 296 |
| Crystal system | Triclinic | Triclinic |
| Space group | $P$-1 | $P$-1 |
| Crystal size (mm) | 0.18 × 0.14 × 0.04 | 0.20 × 0.10 × 0.10 |
| $a$ (Å) | 11.570(2) | 11.347(2) |
| $b$ (Å) | 12.724(2) | 12.553(3) |
| $c$ (Å) | 13.332(2) | 13.485(3) |
| $\alpha$ (°) | 95.177(3) | 95.980(3) |

**Table 3.** *Cont.*

| Complex/Parameters | 1 | 2 |
|:---:|:---:|:---:|
| $\beta$ (°) | 95.600(3) | 94.960(3) |
| $\gamma$ (°) | 90.517(3) | 90.029(3) |
| $V$(Å$^3$) | 1945.0(6) | 1903.2(7) |
| $Z$ | 1 | 1 |
| $D_{\text{calc}}$ (g·cm$^{-3}$) | 1.188 | 1.225 |
| $\mu$ (mm$^{-1}$) | 0.489 | 0.694 |
| $\theta$ range (°) | 1.54–28.70 | 2.26–28.42 |
| $T_{\text{min}}/T_{\text{max}}$ | 0.6407/0.7460 | 0.6300/0.7461 |
| $F$(000) | 750 | 756 |
| Number of parameters | 448 | 472 |
| Reflections collected | 18,697 | 20,250 |
| Unique reflections | 9737 | 9492 |
| Reflections with $I > 2\sigma(I)$ | 5982 | 8310 |
| $R_{\text{int}}$ | 0.0385 | 0.0215 |
| *GooF* | 1.018 | 1.067 |
| $R_1$ ($I > 2\sigma(I)$) | 0.0630 | 0.0332 |
| $wR_2$ ($I > 2\sigma(I)$) | 0.1513 | 0.0870 |
| $R_1$(all data) | 0.1140 | 0.0398 |
| $wR_2$ (all data) | 0.1749 | 0.0904 |
| $\Delta\rho_{\text{min}}/\Delta\rho_{\text{max}}$, e/Å$^3$ | −0.327/0.685 | −0.616/0.428 |

## 5. Summary

Thus, we have synthesized binuclear paddle-wheel complex **1** in which very strong antiferromagnetic coupling takes place, which was shown by experimental methods as well as high-level quantum chemical calculations. The ground spin state of the molecule **1** is $S = 0$, which is indirectly confirmed by the results of magnetic measurements (even with paramagnetic impurity taken into consideration) and EPR spectroscopy. A similar isostructural zinc complex, **2**, and a magnetically diluted sample, **3**, with a cobalt content of about 3.6% were obtained. According to ab initio calculations, easy-plane magnetic anisotropy can be observed in the case of a square-pyramid CoO$_5$ coordination environment with $C_{4v}$ symmetry in the matrix of complexes **1** and **2**. This outcome is consistent with the results of EPR spectroscopy and magnetic susceptibility measurements. The latter are approximated with the assumption of a magnetically isolated ion with $g_{\text{iso}}$ and $D$ parameters with values close to the ones obtained by quantum-chemical calculations. The slow relaxation of the magnetization for sample **3** that is observed under a 1000 Oe external DC-magnetic field proves the perfect isolation of Co(II) ions in the matrix of isostructural Zn(II) complex **2**. The results of magnetic measurement approximation for sample **3** are in good agreement with theoretical calculations, revealing insights into the relaxation mechanism, which is the combination of Raman and QTM processes.

**Supplementary Materials:** The following are available online at https://www.mdpi.com/article/10.3390/cryst14010076/s1, Figure S1—IR spectra, Figure S2—Raman spectra, Figures S3–S5—XPRD data; Figure S6—Calculated splitting of two lowest Kramers doublets; Figure S7—*M*(*H*) data; Figure S8—EPR spectra; Figures S9 and S10—AC magnetic data, Figure S11—Cole–Cole plots; Table S1—Parameters of D-H···O interactions; Table S2—Parameters of C—H···π interactions; Table S3—BS-DFT-calculated energies; details of DFT and CASSCF/NEVPT2/SINGLE_ANISO

calculations. Additional references used in Supplementary Materials: ORCA program package (version 5.0.1) [70]; active space self-consistent field (SA-CASSCF) wave function [71]; N-electron valence second-order perturbation theory (NEVPT2) [72]; the relativistic approximation Douglas–Kroll–Hess (DKH) [73]; polarized triple-ɜ-quality basis set DKH-def2-TZVP [74]; an auxiliary def2/JK Coulomb fitting basis set [75]; quasi-degenerate perturbation theory (QDPT) [76]; the Breit-Pauli form of the spin-orbit coupling operator (SOMF) [75] and an effective Hamiltonian approach [77]; the *ab initio* Ligand Field Theory (AILFT) [78]; Orbach and Raman relaxation process [79,80]; QTM suppressing relaxation in zero field [81].

**Author Contributions:** Conceptualization, validation, M.A.K. and S.A.N.; methodology, T.V.A. and E.N.E.; formal analysis, S.A.N., D.S.Y., E.N.Z.-T. and M.A.K.; investigation, S.N.M. and M.E.N., M.A.S., A.V.K., N.N.E., S.L.V. and A.S.B.; computational study, A.K.M.; writing—original draft preparation, S.A.N.; writing—review and editing, M.A.K. and I.L.E.; supervision, I.L.E. All authors have read and agreed to the published version of the manuscript.

**Funding:** This research was funded by the Ministry of Science and Higher Education of the Russian Federation, grant number 075-15-2020-779.

**Data Availability Statement:** Supplementary crystallographic data for the compounds synthesized are given in CCDC numbers 2320605 (for **1**) and 2320606 (for **2**). These data can be obtained free of charge from The Cambridge Crystallographic Data Centre via www.ccdc.cam.ac.uk/data_request/cif (accessed on 22 December 2023).

**Acknowledgments:** The compounds' characterization was performed using the equipment at the Center for Collective Use of the Kurnakov Institute of General and Inorganic Chemistry of the RAS (X-ray diffraction analysis, ICP-OES, CHN, Raman and IR spectral analyses, magnetochemical studies), which operates with the support of the state assignment of the IGIC RAS in the field of fundamental scientific research.

**Conflicts of Interest:** The authors declare no conflict of interest. The funder had no role in the design of the study; in the collection, analyses, or interpretation of data; in the writing of the manuscript; or in the decision to publish the results.

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
