# Peer review of "Cobalt(II) Paddle-Wheel Complex with 3,5-Di(tert-butyl)-4-hydroxybenzoate Bridges: DFT and ab initio Calculations, Magnetic Dilution, and Magnetic Properties"

_crystals, doi:10.3390/cryst14010076_

Round 1

Reviewer 1 Report

Comments and Suggestions for Authors

Dr. Nikolaevskii and Kiskin present a series of studies on the Cobalt(II) paddle-wheel complex with 3,5-di(tert-butyl)-4-hydroxybenzoate bridges, covering DFT and ab initio calculations, magnetic dilution, and magnetic properties. The content and structure of the article are reasonably complete. However, I have the following opinions on the entire article. After a major revision, the article could be considered for publication:

1.    The introduction by the authors is excessively lengthy. Some irrelevant examples, such as Cu, Mn, V, etc., could be significantly reduced.

2.    In the AC susceptibility section, a Cole-Cole Plot should be provided.

3.    As mentioned by the authors, since the Orbach process does not play a significant role here, an Arrhenius Plot (ln(tau)-1/T) in Fig. 7 might not be entirely appropriate. Using ln(v)-ln(T) would be more suitable.

4.    The formula used in Fig. 7 should be explicitly stated.

5.    To facilitate a thorough examination of the authors' calculations, the ORCA input files for BS/DFT and Ab initio calculations should be included in the supporting information.

6.    Typos should be noted. For example, in line 60, [Co(SPh)4]2-, the "2-" should be an upper script, not a subscript.

Apart from these points, I do not have any further comments.

Author Response

  1. The introduction by the authors is excessively lengthy. Some irrelevant examples, such as Cu, Mn, V, etc., could be significantly reduced.

According to the reviewer’s comment, the introduction has been shortened.

  1. In the AC susceptibility section, a Cole-Cole Plot should be provided.

Thank you for the useful comment. The Cole-Cole Plots were added in ESI as Fig.S11. These data imply a single relaxation magnetic center.

  1. As mentioned by the authors, since the Orbach process does not play a significant role here, an Arrhenius Plot (ln(tau)-1/T) in Fig. 7 might not be entirely appropriate. Using ln(v)-ln(T) would be more suitable.

Thank you for the comment, we used v(T) dependency.

  1. The formula used in Fig. 7 should be explicitly stated.

As it was mentioned in previous version of the manuscript, the data on Fig. 7 were approximated by the sum of QTM (τQТM-1 = B) and Raman (τRaman−1 = CRaman·Tn). The equation used is:

τ−1 = CRaman·Tn + B

In the current version equation used is:

ν = CRaman·Tn

  1. To facilitate a thorough examination of the authors' calculations, the ORCA input files for BS/DFT and Ab initio calculations should be included in the supporting information.

Inputs for calculations were included in ESI

  1. Typos should be noted. For example, in line 60, [Co(SPh)4]2-, the "2-" should be an upper script, not a subscript.

It was corrected and the text was checked.

Reviewer 2 Report

Comments and Suggestions for Authors

In this work, the authors conducted a comprehensive investigation on the new binuclear complex, [Co2(bhbz)4(EtOH)2]·4EtOH (1), and its isostructural zinc counterpart. The molecular structures were determined using single crystal X-ray diffraction. DFT calculations revealed strong Co-Co antiferromagnetic exchange interactions in the binuclear fragment. Substituting one paramagnetic ion with a diamagnetic zinc(II) ion in the binuclear molecule resulted in the remaining cobalt(II) ion acting as an isolated center with magnetic anisotropy.

This work can be interesting for the computational chemistry and the industrial community. As such, the proposed article deserves to be published and the Crystals is certainly well targeted. Before the publication, I would like to ask the authors to consider the minor comments below.

1. page 6, line 214

“at the B3LYP/def2-TZVP level of theory”

It’s preferred to add references for the functional and the basis set here instead of in the SI. B3LYP: Phys. Rev. B 37, 785 and J. Chem. Phys. 1993, 98 (7), 5648–5652. def-TZVP: Phys. Chem. Chem. Phys. 7, 3297.

2. page 6, section 2.3.1. BS/DFT

The long-range interaction can play an important role in the studied systems. Can the authors discuss the effects of not adding the dispersion correction in the DFT calculations?

3. page 6, section 2.3.1. BS/DFT

In the DFT calculations, did the authors use any implicit solvent model to cover the solvation effect?

4. page 8, Figure 3

In this figure, are the energy levels shifted so the dxy orbitals are aligned at 0? I suggest to add the Fermi level information in the figure and in the caption for the better readability.

5. page 9, Figure 5

What does the blue curve stand for? The explanation is needed in the caption.

Comments on the Quality of English Language

No major language or grammar problem found.

Author Response

  1. page 6, line 214

“at the B3LYP/def2-TZVP level of theory”

It’s preferred to add references for the functional and the basis set here instead of in the SI. B3LYP: Phys. Rev. B 37, 785 and J. Chem. Phys. 1993, 98 (7), 5648–5652. def-TZVP: Phys. Chem. Chem. Phys. 7, 3297.

These references were added to the main text.  

  1. page 6, section 2.3.1. BS/DFT

The long-range interaction can play an important role in the studied systems. Can the authors discuss the effects of not adding the dispersion correction in the DFT calculations?

For BS calculations, the geometry of the complex obtained by X-ray data and the corresponding xyz coordinates were taken. The main goal of this article was to study the magnetic interaction between Co2+ ions. Since the complex is considered without optimization and taking into account intermolecular interactions, the use of D3BJ (for example) does not make significant changes to the antiferromagnetic effect.

  1. page 6, section 2.3.1. BS/DFT

In the DFT calculations, did the authors use any implicit solvent model to cover the solvation effect?

Since the geometry was taken from X-ray data, optimization of the complex was not carried out. For BS calculation, taking into account the CPCM correction does not introduce any additional effects on the spin density of Co2+ ions.

  1. page 8, Figure 3

In this figure, are the energy levels shifted so the dxy orbitals are aligned at 0? I suggest to add the Fermi level information in the figure and in the caption for the better readability.

Using AILFT involves calculating the relative energies of d-orbitals in a crystal field with equal weight. And considering the probability of finding electrons in one of the orbitals with respect to the symmetry of the CF. dxy orbitals are at relative energy 0 cm-1, this is noted in the caption to the Figure 3. However, in comparison with the Fermi level (which is rather used in band theory), this is significantly lower.

  1. page 9, Figure 5

What does the blue curve stand for? The explanation is needed in the caption.

It was corrected: Fig. 5. Temperature dependence of χT for 3 (Н = 5 кOe). Blue line is simulation of experimental data in the temperature range 2-130 K (see main text).

Round 2

Reviewer 1 Report

Comments and Suggestions for Authors

The manuscript is suitable for the current edition. I have no more problem.